# High Mutational Heterogeneity, and New Mutations in the Human Coagulation Factor V Gene. Future Perspectives for Factor V Deficiency Using Recombinant and Advanced Therapies

**DOI:** 10.3390/ijms22189705

**Published:** 2021-09-08

**Authors:** Sara Bernal, Irene Pelaez, Laura Alias, Manel Baena, Juan A. De Pablo-Moreno, Luis J. Serrano, M. Dolores Camero, Eduardo F. Tizzano, Ruben Berrueco, Antonio Liras

**Affiliations:** 1Department of Genetics, Santa Creu i Sant Pau Hospital and IIB Sant Pau, 08041 Barcelona, Spain; sbernal@santpau.cat (S.B.); lalias@santpau.cat (L.A.); mbaena@santpau.cat (M.B.); 2CIBERER. U-705, 18014 Barcelona, Spain; 3Department of Pediatric and Oncohematology, University Hospital Virgen de las Nieves, 18014 Granada, Spain; irene.pelaez.sspa@juntadeandalucia.es; 4Department of Genetic, Physiology and Microbiology, School of Biology, Complutense University, 28040 Madrid, Spain; jdepablo@ucm.es (J.A.D.P.-M.); luisserr@ucm.es (L.J.S.); 5Association for the Investigation and Cure of Factor V Deficiency, 23002 Jaén, Spain; lolakamero5@gmail.com; 6Department of Clinical and Molecular Genetics, University Hospital Vall d’Hebron and Medicine Genetics Group, Vall d’Hebron Research Institute, 08035 Barcelona, Spain; etizzano@vhebron.net; 7Pediatric Hematology Department, Hospital Sant Joan de Déu, University of Barcelona and Research Institute Hospital Sant Joan de Déu, 08950 Barcelona, Spain; ruben.berrueco@sjd.es

**Keywords:** factor V deficiency, parahemophilia, Owren’s disease, mutation analysis, advanced therapies

## Abstract

Factor V is an essential clotting factor that plays a key role in the blood coagulation cascade on account of its procoagulant and anticoagulant activity. Eighty percent of circulating factor V is produced in the liver and the remaining 20% originates in the α-granules of platelets. In humans, the factor V gene is about 80 kb in size; it is located on chromosome 1q24.2, and its cDNA is 6914 bp in length. Furthermore, nearly 190 mutations have been reported in the gene. Factor V deficiency is an autosomal recessive coagulation disorder associated with mutations in the factor V gene. This hereditary coagulation disorder is clinically characterized by a heterogeneous spectrum of hemorrhagic manifestations ranging from mucosal or soft-tissue bleeds to potentially fatal hemorrhages. Current treatment of this condition consists in the administration of fresh frozen plasma and platelet concentrates. This article describes the cases of two patients with severe factor V deficiency, and of their parents. A high level of mutational heterogeneity of factor V gene was identified, nonsense mutations, frameshift mutations, missense changes, synonymous sequence variants and intronic changes. These findings prompted the identification of a new mutation in the human factor V gene, designated as *Jaén-1*, which is capable of altering the procoagulant function of factor V. In addition, an update is provided on the prospects for the treatment of factor V deficiency on the basis of yet-to-be-developed recombinant products or advanced gene and cell therapies that could potentially correct this hereditary disorder.

## 1. Introduction

Active factor V (FVa), sometimes referred to as labile factor or proaccelerin, is an essential clotting factor that plays a key role in the blood coagulation cascade on account of its procoagulant and anticoagulant activity [1,2,3]. Eighty percent of circulating FVa is produced in the liver and the remaining 20% originates in the α-granules of platelets [1,4,5]. This means that any disorder affecting the liver will significantly impact the concentration of FVa in plasma and thus disrupt the coagulation process. FVa shares approximately 40% functional and structural homology with coagulation factor VIII (FVIII), except in the B domain region where homology is around 15% [6,7,8]. Both proteins are structurally identical in their A1-A2-B-A3-C1-C2 domains and, in both, the large B-domain is cleaved off the full-length protein as a result of proteolytic activation [7,9].

Circulating FVa is a glycosylated 330kDa polypeptide which loses its B-domain on activation and, as a result, transforms itself into a dimeric molecule made up of a heavy 105 kDa chain that comprises domains A1 and A2 as well as a light 71–74 kDa chain comprising domains A3, C1 and C2. The heavy chain interacts with activated factor X (FXa) and with prothrombin, while the light chain interacts with membrane phospholipids. Activated protein C acts on three arginine residues of the heavy chain, inhibiting the pro-coagulant activity of FVa [10]. The presence of mutations at these cleavage sites gives rise to the so-called FVa Leiden, which creates a state of hypercoagulability [11]. 

Within the coagulation cascade, FVa is a non-enzymatic cofactor of the prothrombinase complex, which accelerates the conversion of prothrombin into thrombin [1]. This enzymatic complex is made up of activated FVa, calcium, phospholipids and FXa. FVa increases the concentration of FXa on the surface of the cell membrane and acts as an FXa co-receptor that allosterically modifies the active site of FXa, which enhances its ability to cleave prothrombin [12]. Figure 1 shows the functioning of the blood clotting mechanism according to Hoffman & Monroe’s now generally accepted cell-based model [13,14] whereby coagulation factors assemble into complexes, whose stability is largely dependent on calcium and the endothelial membranes [15,16]. Following the purification of human plasma-derived factor V from plasma and platelets [17,18], recent efforts have resulted in a human recombinant FVa [19,20,21]. The possibility that a viable FVa deficient animal model may be developed in the medium- to short-term will greatly facilitate research into novel advanced therapies [22,23,24].

In humans, the factor V gene (*F5* gene) is about 80 kb in size and is located on chromosome 1q24.2. The fact that the *F5* and *F8* genes are homologous in terms of their structure and organization suggests that they could be derived from a common ancestor. The coding sequence of *F5* is divided into 25 exons and 24 introns and the length of its cDNA is 6914 bp [25].

The existence of multiple splicing alternatives was described in 2008 [26] and, ever since the first mutation in the *F5* gene was described in 1998 [27], a total of 190 mutations have been reported in the *F5* gene, the most numerous ones being missense and nonsense mutations, followed by small deletions, splicing mutations, small insertions, large insertions, complex rearrangements and a few large deletions. These mutations in the *F5* gene have been identified and subsequently registered in the 1000 Genomes Browser database [28]. The large number of mutations leading to a severe deficiency of FVa confirms the significant allelic heterogeneity of the disease.

Only two large-scale deletions have been reported: an interstitial one in chromosome 1q caused by a drastic reduction of FVa levels to 9%, which produces a nonsense mutation (p.Ser262Trp) in the other *F5* allele [29]; and a heterozygous 205 kb deletion spanning exons one to seven that is accompanied by a splicing mutation (F5 1975 + 5G > A) and results in FVa levels of 1% [30]. The striking scarcity of large-scale deletions in the mutational spectrum of the *F5* gene is probably due to the fact that the gene’s rearrangements cannot be detected using conventional screening methods based on amplification and sequencing of individual exons.

FVa deficiency, also known as parahemophilia or Owren’s disease, is an autosomal recessive coagulation disorder associated with mutations in the *F5* gene [31,32]. Homozygous and compound heterozygous FVa deficiency has been classified as an ultra-rare disease as it occurs in only one to nine per million live births (ORPHA:326) [33], although its incidence is 10 times higher in Western Asian countries where consanguinity is more common [34]. The disease is much less frequent in North America, but it is extremely prevalent in Caucasian countries, which are home to 67% of affected patients [35]. The greater ability to evaluate and detect patients in these countries as opposed to countries with limited health resources could constitute a bias in the calculation of the overall prevalence of the condition.

This hereditary coagulation disorder was first identified in Norway in 1943 by Owren [36], who posited the existence of a fifth essential component in the formation of fibrin, hence the term factor V. This is when the practice began of designating coagulation factors by Roman numerals. Clinically, the disease is characterized by the occurrence of mild to severe bleeding episodes [1]. Platelet-derived FVa is the best predictor of the severity of the condition in terms of circulating levels of FVa [37,38,39], and tissue factor pathway inhibition (TFPI) could act as an additional phenotypic modulator [40]. Consequently, no correlation can be established between FVa levels and the severity of clinical symptoms, although it is likely that the lower the FVa concentration the more severe the symptoms. Although the fastest and simplest procedure to evaluate potential symptoms or bleeding episodes is based on the measurement of plasma FVa levels, it is certainly not the best method to predict the evolution of the disease, which would require consideration of platelet FVa levels. However, plasma FVa measurement-based methods could provide a fairly accurate idea of the patient’s potential clinical symptoms.

As with FVIII and factor IX in hemophilia, plasma levels of FVa determine the severity of the condition in terms of the risk of bleeding episodes and the seriousness of other symptoms. Here too, three levels of severity can be distinguished: mild (>10% of normally circulating FVa), moderate (<10% of normally circulating FVa) or severe (<1% of normally circulating FVa) [1,41,42]. Bleeding episodes usually begin before the age of six and are associated with a heterogeneous spectrum of hemorrhagic manifestations ranging from mucosal or soft tissue bleeds (such as epistaxis or hemarthrosis), to potentially fatal hemorrhages. Abundant nasal and menstrual bleeding are a hallmark in these patients; heavy bleeding is also common during minor and major surgery as well as in the course of dental procedures. Hemorrhagic arthropathy, hematomas and cranial or gastrointestinal hemorrhages are less common [1].

As no plasma-derived or recombinant factor V concentrates are available, current treatment of this condition consists either in administration of, preferably inactivated, fresh frozen plasma (FFP) or in the administration of platelet concentrates [43,44]. Octaplas^®^, a pharmaceutical product treated through a solvent/detergent process, is a good alternative to single donor-derived fresh frozen plasma that exhibits a high safety profile against emerging pathogens [45,46,47]. It is characterized by an optimal quantitative and qualitative composition, which, as a result of its high stability performance, remains unchanged from batch to batch for up to 4 years at a temperature of ≤−18 °C; its high fibrinogen concentration; and controlled levels of the different clotting and fibrinolytic factors and their inhibitors (FVa level ≥ 0.5 IU/mL). Apart from FFP, local and systemic antifibrinolytics such as tranexamic acid can be administered orally or intravenously in order to control bleeding (especially from mucosae) and prior to invasive surgical procedures. In patients with persistent menorrhagia, estrogen-progesterone replacement therapy may be used to reduce menstrual bleeding. Generally speaking, this treatment has not been associated with serious complications such as allergic reactions to plasma components, anaphylaxis or problems derived from perfusion of high levels of FFP. Inhibitors (anti- FVa antibodies) have also been reported, although their occurrence has been rare [48].

This article describes an exhaustive study of two patients with severe FVa deficiency, and of their parents. A high level of mutational heterogeneity of *F5* was identified and nonsense mutations, frameshift mutation, missense changes, synonymous sequence variants and intronic changes were found.

These findings prompted the identification of a new mutation in the human *F5* gene, designated as *Jaén-1*, which can alter the procoagulant function of FVa. The study also includes a comparative phylogenetic analysis between FVa and FVIII given their high degree of genetic homology and that they share biogenetic and coagulation cascade pathways. In addition, an update is provided of the prospects for the treatment of this condition on the basis of yet-to-be-developed recombinant products or advanced gene and cell therapies that could potentially correct this hereditary disorder.

## 2. Results

### 2.1. Clinical Characteristics of the Two Patients with Factor V Deficiency

#### 2.1.1. Case A History

The first patient is a Caucasian 14-year-old girl from Jaén, in the south of Spain, who presented with severe FVa deficiency (FV:C <1%). The deficiency was diagnosed a few days after she was born, following administration of antibiotics to address a urinary infection which triggered mild gastric hemorrhages. Although her hemogram was normal with a normal platelet count, clotting tests showed abnormal values for both coagulation pathways. A determination of coagulation factor levels revealed a severe deficiency of FVa (<1%). The other clotting factors exhibited normal levels. Screening of the patient’s parents showed that both were FVa deficient (mild phenotype). The patient did not experience significant hemorrhagic episodes until she was 8 months old, when she developed a hematoma in her right upper gingiva. As this occurred precisely when her teeth were erupting, she required administration of fresh frozen plasma (FFP), which resolved the hematoma within 24 h. At the age of one, two further bleeding episodes required a FFP transfusion, which immediately resolved the hematoma. Subsequently the patient was diagnosed with an atrial septal defect and was referred to the pediatric cardiology department. Corrective surgery was indicated at the age of two. Prophylactic FFP was administered preoperatively and one week after surgery. No intraoperative complications occurred, and correction of the cardiac defect was achieved. In the next few years, the patient presented only with mild episodes of epistaxis, which coexisted with upper airway infections. Local and oral administration of tranexamic acid resolved the hemorrhage in all but one of the cases, in which FFP had to be administered to address the epistaxis. At the age of six she presented with left hip pain of 12 hours’ evolution. A diagnostic ultrasonogram revealed hemarthrosis in the left hip and no other previous trauma. Administration of FFP every 12 h during the first 24 h and every 24 h thereafter over 5 days resulted in progressive improvement. At age seven, she was admitted with vomiting, abdominal pain and mild yet progressive anemization. While in hospital, the patient experienced headache. Her family explained that on her birthday she had sustained a fall leading to mild cranial trauma without loss of consciousness. A cranial CT-scan revealed a predominantly dense extra-axial collection in the left frontal region extending to the cerebellar tentorium arising from a left subdural hematoma with a biconvex anterior portion, which suggested the possibility of an associated epidural component. Following examination by the neurosurgical department, it was decided to correct her coagulation disorder and adopt a watch-and-wait approach. Administration of FFP every 12 h for 5 days resulted in a slight resorption of the hematoma. The patient is now 14 years old and in the past 2 years has presented with various bleeding episodes, some of which required admission to hospital and administration of FFP and Octaplas^®^ (14 mL/kg), which was initiated around 3 years ago.

#### 2.1.2. Case B History

A 2-year-old Pakistani girl was admitted with cellulitis in the context of a severe FVa deficiency. There was a history of consanguinity in the family as her parents were cousins and had been previously diagnosed with mild FVa deficiency in Pakistan. She had two siblings with very mild FVa deficiency but no severe bleeding symptoms. At the age of 10 months, she had presented with a spontaneous nose-bleeding episode after which she was diagnosed with severe FVa deficiency. Up to her arrival in Spain, she had received FFP four times due to mucocutaneous bleeding episodes (two of them secondary to trauma). The child consulted to the hospital due to a suppurative lesion in her left leg and fever. There was a mosquito bite in that area, sustained three days before. A physical examination revealed an ulcerative, suppurative lesion with perilesional erythema and inflammatory symptoms. Blood cultures showed leukocytosis and neutrophilia; prothrombin time was 50.1 s, prothrombin activity was 10.7%, aPTT was 129.5 s and FVa expression levels were 0.7% of normal. The rest of the clotting factors were present at normal levels. Prior to wound debridement surgery, administration of FFP, antifibrinolytic and intravenous antibiotic therapy was determined. As blood cultures obtained at 48 h showed a methicillin-resistant staphylococcus aureus infection, tailored antibiotic therapy with intravenous clindamycin was initiated with excellent results. No bleeding problems were observed during admission, with no further FFP administration being necessary. The drainage was withdrawn after seven days. The patient was discharged on a four-week regimen of oral antibiotic therapy. At present, the patient is almost 3 years old and no further bleeding episodes have been reported.

### 2.2. Hematological and Coagulation Analysis

All hematological parameters in both the patients and their parents, including hemoglobin (Hb) concentration, mean corpuscular hemoglobin (MCH), mean corpuscular hemoglobin concentration (MCHC), total erythrocyte count (RBC), hematocrit (HCT) level, mean corpuscular volume (MCV), reticulocyte (RETIC), total white blood cell (WBC) count, WBC differential count, platelet (PLT) count, mean platelet volume (MPV), platelet distribution widths (PDW) and plateletcrit (PCT), were within the reference ranges.

A coagulation analysis (Table 1) showed altered values for both extrinsic and intrinsic coagulation pathways (prothrombin time, prothrombin activity, activated partial thromboplastin time (cephalin time)), the international normalized ratio, and FVa activity in both patients. Both presented severe FVa deficiency, with FVa activity under 1% of the reference value. In both patients’ parents, alterations in the coagulation parameters and FVa activity were correlated with the type of mutation of the *F5* gene allele. The parents’ heterozygous state induced a mild phenotype.

### 2.3. Molecular Study of the F5 Gene in the Two Patients and Their Parents

All deleterious mutations and genetic variants were assigned a nucleotide number starting at the first translated base of the *F5* gene according to reference sequence *NM_000130.4*. The variant sequences were termed in accordance with the recommendations of the Human Genetic Variant Society (HGVS). Mutation screening of the *F5* gene was carried out in the two girls with a severe FVa deficiency. A total of 24 different sequence variants were identified in this study including 2 nonsense mutations, 1 frameshift mutation, 6 missense changes, 8 synonymous sequence variants and 7 intronic changes (Table 2). Two of them were identified in patient A, as nucleotide substitutions in the exon 13 of the *F5* gene involving an amino acid change (arginine or tryptophan) to a stop codon in the factor V protein: *NM_000130.4: c. 2218C>T* (*p.Arg740**) and *NM_000130.4: c.3279G>A* (*p.Trp1093**). The mutation found in patient B (*NM_000130.4: c.2862del*), also located in exon 13 of the *F5* gene, was a thymine nucleotide deletion. This deletion generated a frameshift mutation leading to an amino acid change from serine to alanine at position 955 in the protein encoded by the *F5* gene, resulting in the appearance of a premature stop codon (*p.Ser955Alafs*4*).

Seventeen out of the 24 variants have a minor allele frequency (MAF) ≥ 5% as described in the above mentioned 1000 Genomes Browser database (5 missense, 7 synonymous and 5 intronic variants) (Table 2). The remaining seven changes do not have a MAF value because they are rare sequence variants and have not been described previously (1 missense, 2 synonymous sequence variants and 2 intronic changes). Figure 2 and Figure 3 show the genetic map of patient A’s *F5* gene and the location of mutations Arg740* and Trp1093*. Figure 4 shows the genetic map of patient B’s FVa gene and the location of mutation 2862del. Ser955Alafs*4.

In addition, a familial segregation pattern was detected in the mutations identified in patient A (Figure 5A). In this patient, both nonsense mutations (*p.Arg740** and *p.Trp1093**) were in a compound heterozygous state. One of them was inherited from her mother (*NM_000130.4: c. 2218C>T*, *p.Arg740**) and the other from her father (*NM_000130.4: c. 3279G>A, p.Trp1093**). Thus, each of her parents have one of these nonsense mutations in heterozygous state. In patient B, the frameshift mutation in the *F5* gene (*NM_000130.4: c.2862del, p.Ser955Alafs*4*) is in homozygous state (Figure 5B). Both her brother and her parents carried this mutation in heterozygous state.

The type of mutation was correlated to FVa activity levels in both the patients and their parents. Mutation *c.3279G>A* in the father of patient A was related to FVa activity of 21% of normal; mutation *c. 2218C>T* in the mother of patient A was associated with FVa activity of 62.9% of normal; and mutation *c.2862del* in both parents of patient B exhibited a FVa activity of, approximately, 45% of normal.

Phylogenetic studies of different vertebrate species performed as part of this analysis showed that the three mutations described in the two patients are all located in well-conserved regions of exon 13 within the B domain of FVa (Figure 6). The zebrafish exhibits more differences than other species in their amino acid sequence probably due to the fact that, phylogenetically, the species is quite distant from humans.

## 3. Discussion

Patients in this study suffered from severe FVa deficiency (<1% of normal factor levels), caused by mutations in both of the *F5* gene’s alleles giving rise to the appearance of a premature stop codon. In the case of patient A, a compound heterozygous situation occurred, with one mutation in each allele, one inherited from the father and the other from the mother. These mutations induced a premature arrest in the synthesis of FVa. In the case of patient B, a homozygous situation occurred carrying the same mutation. 

Of the 24 changes found in the *F5* gene mutation screening procedure carried out in the two girls affected with severe FVa deficiency, three were interpreted as pathogenic mutations (2 nonsense and 1 frameshift) because they met the pathogenicity criteria (the occurrence of a stop codon is in itself a pathogenic trait).

According to in silico studies with the Alamut visual v.2.6 software, all the missense mutations identified in this study should result in a neutral effect. MAF values ≥ 5% and the benign effect associated with the mutations in the *F5* gene were two important criteria that reinforced the idea that 19 of these 24 changes could be considered benign polymorphisms, only the remaining five changes being associated to an unknown clinical effect.

As regards familial segregation, patient A’s mother was heterozygous for the already studied c.2218C>T, p.Arg740* mutation [49] with FVa plasma levels of 63%, indicative of a mild phenotype with no noteworthy clinical symptoms. The father, for his part, was heterozygous for the not-previously-reported c.3279G>A, p.Trp1093* mutation, and presented with FVa plasma levels of 21%, also pertaining to a mild phenotype without significant symptoms.

As regards the c.2218C>T, p.Arg740* mutation observed in patient A, it was first characterized by Lunghi et al. [49] in a patient with FV Leiden. This mutation results in a truncated non-functional protein lacking the full light chain (A3, C1 and C2 domains). Lunghi et al. observed the coexistence of the R506Q mutation, which gives rise to the Leiden variant and to hypercoagulability, and the c.2218C>T, p.Arg740* mutation, which produces a deficiency of functional FVa. Although the latter mutation could in some cases compensate for the Leiden phenotype, the final result is usually plasma hypercoagulability with a significant resistance to APC. Our present study could be said to be the first to document the c.2218C>T, p.Arg740* mutation in a patient with a congenital FVa deficiency.

As far as the c.3279G>A, p.Trp1093* mutation is concerned (the second mutation exhibited by patient A), it is a new, not-yet-reported mutation, which results in a non-functional protein and in a congenital FVa deficiency. The mutation is currently being analyzed in our laboratory for possible correction by advanced therapies and gene editing using CRISPR/Cas9. Both the c.3279G>A, p.Trp1093* mutation and the CRISPR/Cas9 technology, developed to treat coagulopathies arising from this mutation, are protected by a Spanish patent (Ref. ES2785323B2) with international coverage [50,51].

Patient B’s parents were heterozygous for the same mutation (the previously described c.2862del, p.Ser955Alafs*4) because there was a high degree of consanguinity between them [52]. They exhibited FVa plasma levels of 46% without any significant symptoms. Although both her brother and her parents carried this mutation in heterozygous state, her older sister was negative for this mutation in the *F5* gene.

Transmission of the FV deficiency in patient B is marked by very strong consanguinity as she is homozygous for the same mutation inherited from her parents who share a close family connection. In contrast, patient A presents with two different mutations inherited from her parents, each of whom comes from a close endogamous community. Homozygosis, particularly in the case of rare or ultra-rare diseases, generally occurs in countries in Northern Africa, Sub-Saharan Africa, the Arab world, Asia and especially India and Pakistan, where the culture favors consanguineous marriages for social, economic or political reasons. In India and Pakistan, consanguineous marriages account for 5–60% and for more than 73% of total, respectively [53,54,55]. According to some social science studies, there could be a direct relationship between a high rate of consanguinity and a higher incidence of hereditary and disabling conditions [53].

As shown by Naderi et al. [56], this circumstance is particularly striking in the case of FVa deficiency. These authors analyzed 23 patients with FVa deficiency in the Iranian Sistan and Baluchestan provinces, home to a population of 2.7 million and an incidence of FVa deficiency of one in every 50,000 live births, far above the global incidence of the condition, which is one to nine per million live births. Patients exhibited an overall 91.3% consanguinity rate; of them, 66.6% presented with second degree consanguinity and 33.3% were related by first-degree consanguinity. Although consanguinity may have a negative connotation from a healthcare perspective as compared with other hereditary conditions, the fact is that there are over one million people worldwide living in societies where consanguineous marriages are common. In a study of a wide sample of pregnant women, Raj et al. [57] discussed the benefits that consanguinity could entail considering the WHO’s definition of health as a state of complete physical, mental and social wellbeing and not just the notion of absence of disease. In cases like that of patient B, consanguinity seems to have permitted a less pathological mutation in homozygosis than that observed in patient A.

For the reasons above, the screening of mutations in certain populations and regions is sometimes extremely challenging, as most mutations causing FVa deficiency have been identified in single families (“private mutations”). Only the Tyr1702Cys mutation has shown itself to be recurring, at least among the Italian population [58].

An interesting aspect of FVa deficiency, which separates it from other congenital coagulopathies like hemophilia, is the lack of a clear correlation between the disease’s phenotype and its genotype; that is, between the clinical expression of the disease and the plasma levels of FVa and the mutation associated with the condition [59]. In fact, some patients present with a milder (or even nonexistent) hemorrhagic phenotype even if they carry the same mutation, or if they show similar FVa plasma levels. Although both patients in the present study had FVa levels below 1% (severe phenotype) and mutations resulting in a non-functional codon producing a truncated protein, the manifestation (phenotype) of the disease seemed more severe in the case of patient A, as the medical history seems to indicate with more frequent, and sometimes more severe hemorrhagic events. A potential explanation for the difference between both patients may have to do with the above-mentioned consanguinity factor, although it could also be attributed to the co-inheritance of FVa deficient alleles with risk factors for thrombophilia and/or with alleles of the modifier genes, which can influence the overall coagulation cascade [60].

This study has also demonstrated the high heterogeneity of the mutations affecting the *F5* gene. This finding warrants performing a full mutational screening of the gene to obtain a molecular diagnosis of the disease. This wide range of mutational changes observed in the *F5* gene is related to a large number of highly variable effects at a functional and molecular level. Some of the consequences of the different mutations include alterations in the stability of FVa [61]; alterations in the splicing [62,63] and in the biogenesis of FVa [64,65,66,67], mainly due to secretion disturbances; and mutations affecting the binding sites of the protein to certain activation or inactivation factors [68]. The mutations described in this study are related to the appearance of the stop codon, which results in a non-functional truncated protein. An unequal distribution is also observed among exons, with the largest number of mutations –both pathogenic and otherwise– corresponding to exon 13.

From a phylogenetic standpoint, the fact that two proteins share a common biogenetic pathway might indicate that the two molecules could present with similar characteristics. This could for example be the case of FVa and FVIII, both of which are encoded by very large genes and share a high percentage of sequence identity—approximately 40% in the A domain and 35–43% in the C domain [69,70]. The B domain is the one characterized by the lowest level of homology and, evolutionarily, the least preserved one. As a result, the proteins are very large (containing over 2000 amino acids each) and have a very similar domain structure, with the A1–A2 domains forming the heavy chain, the B domain exhibiting a large number of postranslational modifications, and the A3, C1 and C2 domains forming the light chain [71].

Similarities also exist in terms of functionality, which is in line with the phylogenetic evolution hypothesis as far as the conservation of these two proteins is concerned. In both cases, the mechanism of activation is mediated by thrombin, which binds to arginine residues residing in the B domain and leading to its elimination [71,72]. As regards the inactivation of FVa and FVIIIa, the process is identical in both cases and based on the cleavage and elimination of part of the A2 domain of the proteins by APC. Moreover, there is a functional interaction between both coagulation factors as FVa boosts the inactivation efficacy of FVIIIa [71,72,73]. For these reasons, it is not unconceivable that these two proteins may share one same biogenetic transport pathway, including a transport protein (LMAN1) and its cofactor (MCFD2) [74,75].

Genetic homology refers to the relationship between two genes that share a common ancestor; orthologous genes are homologous genes that diverged after a speciation event, and paralogous genes are homologous genes separated by a gene duplication event [76]. The degrees of conservation observed in the different regions of FVa analyzed in our study are in line with the findings of other authors [77,78], who have shown that the B domain tends to be the least conserved one, which is related to its importance in the function of FVa. Although the sequence alignment analyses indicate a lower homology in the case of the zebrafish, it must be noted that that is one of the teleost species where a greater number of orthologous genes are involved in hemostasis [79,80]. That is the reason why zebrafish has been postulated as a useful model for in vivo coagulation studies [81,82].

FVa deficiency is an ultra-rare disease, which has implications both in the social and the healthcare sphere. Indeed, research into novel treatments and diagnostic methods is a challenge for both the scientific community and for society, which are often mutually dependent. On the one hand every patient with a rare or an ultra-rare disease is considered unique and a rich source of valuable information, but on the other they are often neglected because research into these diseases seldom makes economic sense. This is the situation of patients with FVa deficiency, for whom no specific treatment is currently available [83]. This differs from the situation of patients with hemophilia for whom treatment with exogenous coagulation factors has been used for some time [84]. Current treatment of FVa deficiency is based on the administration of fresh frozen plasma, platelet concentrates and a pool of different coagulation factors.

Recombinant factors VIII and IX are nowadays the treatment of choice for hemophilia A and B, respectively [85]. Work is at present underway to develop a recombinant FVa, which will hopefully be available in the medium term for treating a wide range of hemorrhagic disorders, including FVa deficiency, although it is still in the early phases of preclinical development [19,20]. Recombinant DNA methods are safe and highly effective in alleviating the hemorrhagic symptoms and preventing the disabling articular effects experienced by patients. 

Against this background, advanced gene and cell therapies are now able to offer a “curative,” rather than merely substitutive treatment, as they are capable of correcting the mutational root cause of many conditions, particularly congenital ones, both monogenic and polygenic [86,87,88,89] Congenital coagulopathies, many of them rare diseases like hemophilia or ultra-rare diseases like FVa deficiency, may benefit from these developments.

The new advanced therapies are part of a new concept of customized or individualized pharmacology. At present, individualized pharmacology is gaining significant momentum in the treatment of many conditions as it increases the effectiveness of treatments, improves the patients’ quality of life and contributes to a more rational use of medications, which leads to cost savings. With advanced therapies it is not only possible to individualize treatments when the products are already on the market but also during the design phase. This means that the therapeutic methodology best suited to a given patient can be selected according to the patient’s clinical characteristics and to a very rigorous risk/benefit assessment. In fact, pharmacogenomics uses genomic information combined with other “omic” data to individualize the selection and use of medicines with a view to preventing adverse reactions and maximize their efficacy [90].

It would be ideal to develop a treatment offering maximum safety and efficacy. In the realm of advanced therapies, particularly gene therapy, increased efficacy is usually associated with a higher incidence of potential adverse events. Viral vectors [91], which are highly effective due to their total or partial ability to integrate into the genome, are prone to a high rate of adverse events. Conversely, non-viral vectors are extremely safe but not as effective as they rarely allow for therapeutic levels of the expressed protein to be achieved [92,93]. One could also classify cell therapy protocols as a function of the target cells used. Thus, pluripotent cells like embryonic stem cells or induced pluripotent stem cells [94] are the most effective, but at the same time the ones prone to a higher tumorigenicity risk, while mesenchymal stem cells for example, which could be somewhat less effective, have been shown to offer the highest levels of patient safety [95].

Very few cell-therapy approaches have been described in the context of congenital coagulopathies [96]. Our laboratory has obtained highly promising results with transplantable FVa-producing hepatocytes obtained from mesenchymal stem cells (results pending editorial acceptance).

The greatest strides have been taken in the field of gene therapy, where partially integrative adeno-associated viral vectors have been successfully used to treat hemophilia A and B [97]. However, no advanced-therapy approach has as yet been proposed to address other congenital coagulopathies. Studies are being conducted by our laboratory, using CRISPR/Cas9 technology, aimed at the correction of the mutations resulting in FVa deficiency (the results are at an advanced stage of development).

As for other types of therapeutic strategy, it is important to analyze their risk/benefit ratio. In the case of advanced (cell and gene) therapies, this criterion must be particularly demanding as their potential long-term adverse events are not known. Consequently, the inclusion of patients in a clinical trial is contingent on a series of factors. It is not the same to have an optimal substitutive treatment, such as that available for patients with hemophilia A and B, as it is to have no treatment whatsoever as in the case of FVa deficiency. A compromise must be struck based on obtaining the greatest benefit at the lowest risk, considering not only whether a treatment is available or not, but taking into account the phenotype of the disease and the FVa expression levels and expression times sought. In the case of congenital coagulopathies, the success criteria are not as stringent as for other conditions. Indeed, converting patients from a severe to a moderate or mild phenotype should by itself be considered a successful result. This largely conditions the use of gene or cell therapies and viral or non-viral vectors. 

## 4. Materials and Methods

### 4.1. Blood Collection

After the parents signed the informed consent documents, blood samples were drawn from both patients and their parents and placed in tubes containing 0.105 M sodium citrate (1:9 volume). Plasma was obtained following centrifugation at 2000× *g* for 20 min, and subsequently stored in aliquots at −80 °C until further analysis.

### 4.2. Hematological and Coagulation Tests

Hematimetric parameters included: hemoglobin (Hb) concentration, mean corpuscular hemoglobin (MCH), mean corpuscular hemoglobin concentration (MCHC), total erythrocyte count (RBC), hematocrit (HCT) level, mean corpuscular volume (MCV), reticulocyte (RETIC), total white blood cell (WBC) count, WBC differential count, platelet (PLT) count, mean platelet volume (MPV), platelet distribution widths (PDW) and plateletcrit (PCT). All of these parameters were obtained in an automated cell counter (ADVIA^®^120 Hematology System; Siemens Healthcare GmbH, Zurich, Switzerland). Blood samples for the analysis of platelet function and bleeding times were collected in evacuated tubes (Vacutainer, Becton Dickinson & Co, Franklin Lakes, NJ, USA), containing 3.8% citrate and processed in a Platelet Function Analyzer-100 (PFA-100^®^; Siemens Healthcare Diagnostics AG, Zurich, Switzerland) as described by Parri [98]. Collagen/adenosine-5-diphosphate (CAPD) cartridges (Dade^®^ PFA Collagen/ADP Test Cartridge, Siemens Healthcare Diagnostics, AG, Zurich, Switzerland) were brought to room temperature and approximately 1 mL of the total citrated blood was added. Samples were aspirated under constant vacuum from a reservoir through a capillary and a microscopic opening in a collagen- and ADP-coated membrane. The PFA-100 test was used to measure the time needed for occlusion of the aperture by platelet plug formation.

All hemostasis-related parameters such as prothrombin time (PT), prothrombin activity (PA), activated partial thromboplastin time (aPTT) (cephalin time), fibrinogen (F) and international normalized ratio (INR), were determined in accordance with the standards defined by the Hematology Department of the Jaen University Hospital, Spain. HemosIL^®^ kits were used (Instrumentation Laboratory; Bedford, MA, USA), as per the manufacturer’s instructions.

HemosIL^®^ RecombiPlasTin 2G was used to measure prothrombin time (PT) and fibrinogen levels as described by Tripodi [99]. A high-sensitivity thromboplastin reagent was used, based on recombinant human tissue factor (rhTF). Following reconstitution with RecombiPlasTin 2G diluents, the PT reagent included in the RecombiPlasTin 2G kit turned into a liposomal preparation containing rhTF relipidated in a synthetic phospholipid blend and combined with calcium chloride, buffer and a preservative. As rhTF does not contain any contaminating coagulation factor, RecombiPlasTin 2G is highly sensitive to extrinsic pathway coagulation factors, which makes it particularly suited to assays with such factors. The formulation of RecombiPlasTin 2G makes it insensitive to therapeutic heparin levels. In the PT test, addition of the reagent to the patient’s plasma in the presence of calcium ions activates the extrinsic coagulation pathway. This eventually results in the conversion of fibrinogen into fibrin, with the formation of a solid gel. Fibrinogen must be quantitated (PT-based method) by relating the absorbance, or light scatter, during clot formation, to a calibrator. PT results are reported in seconds, activity percentage, or INR; fibrinogen levels, in g/L.

HemosIL^®^ APTT-SP (liquid) was used to determine the aPTT value, as described by Van de Bresselaar [100,101]. The aPTT test utilizes a contact activator (a colloidal silica dispersion with synthetic phospholipids, buffer and a preservative) that stimulates factor XIIa production. This activator provides a contact surface for the interaction of high molecular weight kininogen, kallikrein and factor XIIa. Such contact-based activation occurs at 37 °C over a given period of time. Addition of 0.025 M calcium chloride with a preservative triggers a series of reactions that will result in clot formation. Phospholipids are required to generate the compounds that will activate factor X and prothrombin. Results are expressed in seconds.

HemosIL^®^ FVa deficient plasma was used for the quantitative determination of FVa activity, as described by Tripodi [99]. FVa activity in plasma was determined using the modified prothrombin time test conducted in the patient’s citrated diluted plasma in the presence of FVa immunodepleted human plasma (≤1%). Correction of prolonged coagulation time for the deficient plasma was proportional to the concentration (activity percentage) of the specific factor (FVa) in the patient’s plasma, which can be obtained by plotting a calibration curve. FVa reference levels ranged between 60 and 130% (0.60–1.30 IU).

### 4.3. DNA Extraction

Following informed consent, EDTA anticoagulated blood samples were collected. Genomic DNA was automatically extracted from peripheral leukocytes using the salting-out procedure (Qiagen, Düsseldorf, Germany).

### 4.4. Mutational Analysis

Primers corresponding to the complete coding sequence and intronic adjacent regions of the F5 gene were designed according to entry NC_000001.11 of the NCBI database to yield PCR products in a range between 300 and 600 bp approximately (Table 3). PCR amplification was performed in a final volume of 25 μL that contained 1.5 mM–2.5 mM MgCl_2_ (depending on the fragment), 200 μM of each dNTP, 0.2 μM of each primer, 0.5 units of Taq DNA polymerase (Ecogen, Madrid, Spain) in the recommended buffer and 150–200 ng of genomic DNA. The thermocycling conditions used were as follows: 94 °C for 5 min, followed by 30 cycles at 94 °C for 30 s (denaturation step), 58–60 °C (depending on the fragment amplified) for 30 s and 72 °C for 2 min, followed by a 20 min final extension step at 72 °C. When PCR amplification products were completed, any inconsumable dNTPs and primers remaining in the PCR product mixture were removed by the action of the ExoSAP digestion enzymes (exonuclease I and shrimp alkaline phosphatase). After the clean-up procedure was completed, the bidirectional sequencing reaction (forward and reverse primers) was performed by the Sanger method using BigDye v.1.1. terminator chemistry. The sequencing reactions of the fragments amplified were cleaned by a Montage SEQ96 kit on a vacuum manifold, where the unincorporated dye terminators and salts were removed from sequencing products prior to their subjection to capillary electrophoresis on an ABI Prism 3500 genetic analyzer (Applied Biosystems, Massachusetts, USA). Sequence chromatograms were analyzed using SeqScape v.2.1.1. Mutations and genetic variants were described according to the Human Genome Variation Society (HGVS) nomenclature [102]. All nucleotide changes identified in the patients with FVa deficiency were analyzed by Alamut visual v.2.6. software (Sophia Genetics, Inc, Boston, USA), which integrates genetic and genomic information from different sources into one consistent and convenient environment to describe variants using HGVS nomenclature and help interpret their pathogenic status. Furthermore, all changes detected in the patients were studied in both their parents to determine family segregation.

### 4.5. Sequence Alignment

A multiple alignment of protein sequences from multiple vertebrate animal species (*Canis lupus familiaris*, *Oryctolagus cuniculus**, M**us musculus, Rattus norvegicus* and *Danio rerio*) was carried out using the T-Coffee alignment tool [103], which combines different sequence-alignment methods to find out whether the mutations described are structurally and functionally important and whether they are located in conserved regions.

## Figures and Tables

**Figure 1 ijms-22-09705-f001:**
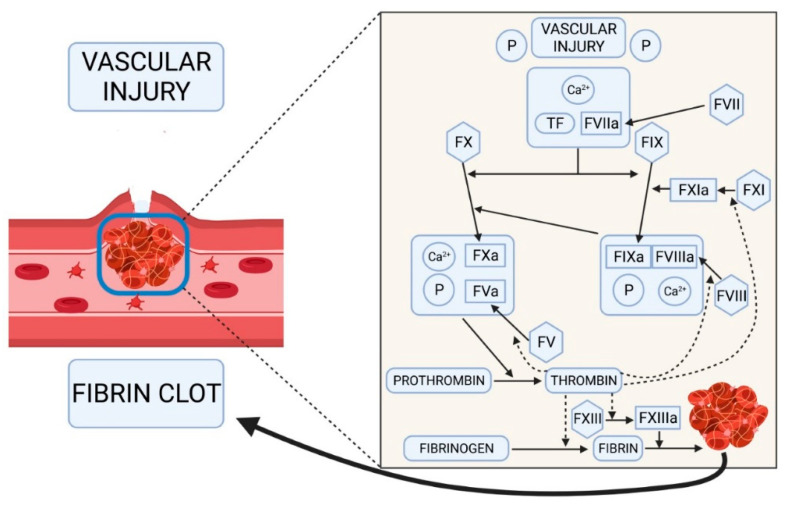
Blood clotting mechanism. Hoffman & Monroe’s cell-based model of hemostasis according to which coagulation factors form multimolecular complexes, calcium and the endothelial membranes playing a key role in the stabilization of such complexes. Abbreviations: P, membrane phospholipids; TF, tissue factor; F, factor. Created in Biorender.com (accessed on 6 September 2021).

**Figure 2 ijms-22-09705-f002:**
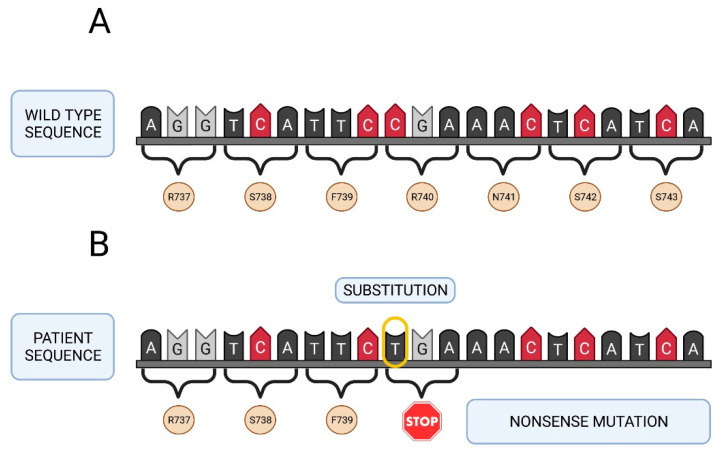
Genetic map of the *F5* gene of patient A. (**A**) Native sequence. (**B**) Sequence of the patient with the Arg740* mutation (premature stop codon, nonsense mutation). Created in Biorender.com (accessed on 6 September 2021).

**Figure 3 ijms-22-09705-f003:**
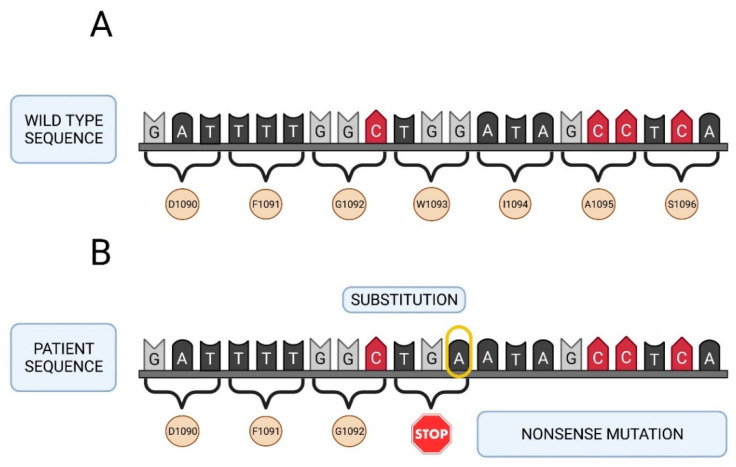
Genetic map of the *F5* gene of patient A. (**A**) Native sequence. (**B**) Sequence of the patient with the Trp1093* mutation (premature stop codon, nonsense mutation). Created in Biorender.com (accessed on 6 September 2021).

**Figure 4 ijms-22-09705-f004:**
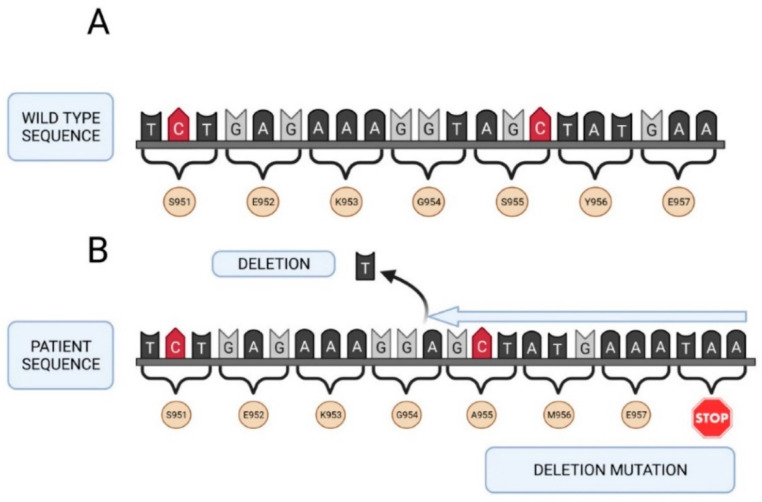
Genetic map of the *F5* gene of patient B. (**A**) Native sequence. (**B**) Sequence of the patient with the 2862del, Ser955Alafs*4 mutation (premature stop codon, deletion mutation). Created in Biorender.com (accessed on 6 September 2021).

**Figure 5 ijms-22-09705-f005:**
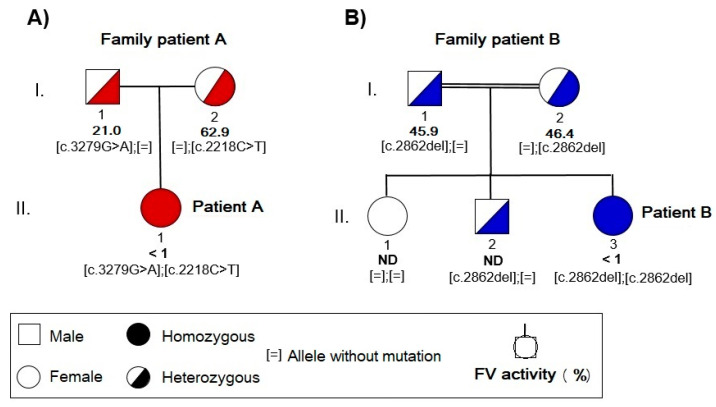
Family pedigrees of the FVa deficient Spanish and Pakistani families. FVa levels and segregation of the identified mutations are shown. (**A**) Patient A with FVa deficiency and two mutations in compound heterozygous state with familiar segregation (*c.3279G>A* mutation inherited from her father and *c.2218C>T* from her mother). (**B**) Patient B with FVa deficiency and mutation in homozygous state, the *c.2862del* mutation inherited from both, father and mother.

**Figure 6 ijms-22-09705-f006:**
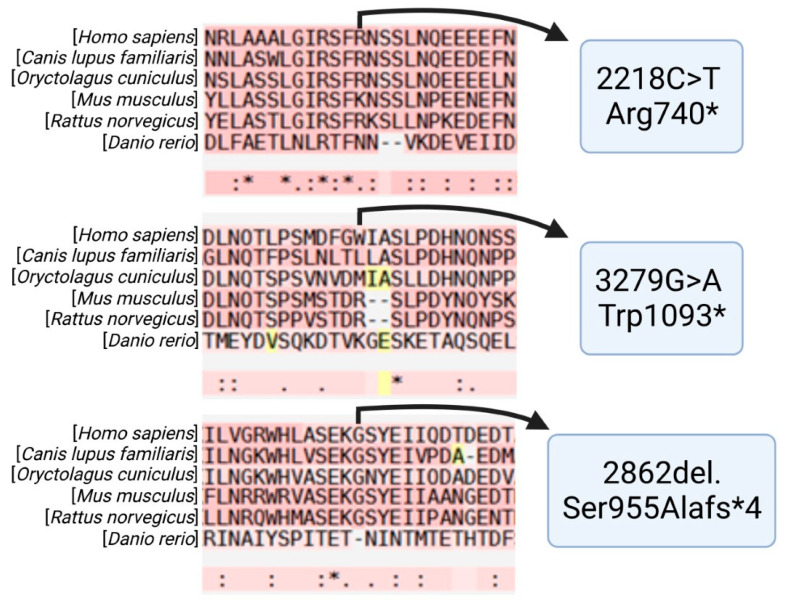
Multiple sequence alignment comparing the different regions of FVa corresponding to the 2218C>T Arg740, 3279G>A Trp1093* and 2862del, Ser955Alafs*4 mutations in *Homo sapiens*, *Canis lupus familiaris*, *Oryctolagus cuniculus*, *Mus musculus*, *Rattus norvegicus* and *Danio rerio*. The mutated amino acid is at the center, with an arrow pointing to it. The three mutations described in the two patients are in exon 13, corresponding to the protein’s B domain, and in well-conserved regions across the different vertebrate species analyzed. Created in Biorender.com (accessed on 6 September 2021).

**Table 1 ijms-22-09705-t001:** Coagulation parameters and factor V activity.

	PT	PA	aPTT	F	INR	FVa
Patient A *	49.6	16.0	146.5	4.4	4.6	<1
Patient B **	50.1	10.7	129.5	3.7	5.8	<1
Father A	12.2	83.0	37.8	3.2	1.1	21.0
Mother A	11.0	96.0	34.4	3.6	1.0	62.9
Father B	11.5	86.8	30.3	3.3	1.1	45.9
Mother B	12.0	92.3	26.8	3.7	1.0	46.4

* Patient from Jaén, Spain. ** Patient from Pakistan. PT, prothrombin time (8–13 s); PA, prothrombin activity (75–120%); aPTT, activated partial thromboplastin time (cephalin time) (24–37 s); F, fibrinogen (2–6 g/L); INR, international normalized ratio (0.8–1.2); and FVa, factor V activity (60–130% or 0.6–1.3 IU).

**Table 2 ijms-22-09705-t002:** Genetic variants identified in the *F5* gene in patient A.

	F5 Gene ^†^	Exon	Intron	FV Protein	SNP	MAF ^§^	Polyphen	M-Taster	SIFT
1	c.237A>G	2		*p.Gln79Gln*	rs6028	23	**	**	**
2	c.952+76C>T		6	*	rs2239854	27	**	**	**
3	c.1238T>C	8		*p.Met413Thr*	rs6033	5	Benign	Polymorphism	Permissive
4	c.1242A>G	8		*p.Lys414Lys*	rs6035	7	**	**	**
5	c.1297-121T>C		8	*	rsl0800456	43	**	**	**
6	c.1380C>T	9		*p.Asn460Asn*	rs6015	5	**	**	**
7	c.1397-164C>T		9	*	rs7537742	50	**	**	**
8	c.1716G>A	11		*p.Glu572Glu*	rs6036	5	**	**	**
9	c.1926C>A	12		*p.Thr642Thr*	rs6037	5	**	**	**
10	**c.2218C>T**	**13**		** *p.Arg740 ** **	******	******	******	******	******
11	c.2289A>G	13		*p.Glu763Glu*	rs6024	5	**	**	**
12	c.2450A>C	13		*p.Asn817Thr*	rs6018	5	Benign	Polymorphism	Permissive
13	c.2758A>G	13		*p.Arg920Gly*	**	**	Benign	Polymorphism	Deleterious
14	**c.3279G>A**	**13**		** *p.Trp1093 ** **	******	******	******	******	******
15	c.3865T>C	13		*p.Phe1289Leu*	**	**	Benign	Polymorphism	Permissive
16	c.3939C>T	13		*p.Ser1313Ser*	**	**	**	**	**
17	c.3980A>G	13		*p.His1327Arg*	rs1800595	5	Benign	Polymorphism	Permissive
18	c.4095C>T	13		*p.Thr1365Thr*	rs9332607	26	**	**	**
19	c.4796+50A>C		13	*	**	**	**	**	**
20	c.5209-134T>C		15	*	**	**	**	**	**
21	c.5290A>G	16		*p.Met1764Val*	rs6030	29	Benign	Polymorphism	Permissive
22	c.6194-20C>A		22	*	rs6013	5	**	**	**
23	c.6529-65A>C		24	*	rs2227243	5	**	**	**
24	c.6665A>G	25		*p.Asp2222Gly*	rs6027	5	Benign	Polymorphism	Permissive

† The nomenclature follows the recommendations of the Human Genome Variant Society (HGVS). The cDNA reference sequence of the *F5* gene is *NM_000130.4.*
**^§^** MAF: minor allele frequency obtained from dbSNP data base. The predictions were made in silico with the *Alamut visual v.2.6 software* that uses the prediction programs Polyphen, Mutation Taster and SIFT. * Not applicable. ** No data available.

**Table 3 ijms-22-09705-t003:** Primers design for human *F5* analysis.

Exon *	Primer	Sequence 5′ → 3′	Size (bp)
1	F5_1F	CACCTGCAGTAA AACAGTCAC	532
	F5_1R	AGCCATGACATTGCAAAGGG	
2	F5_2F	ACAGTTTGGGTTTCTACTGTG	461
	F5_2R	GCATGTGAATGCCAAATTACCC	
3	F5_3F	AAGTGAGTCAGCCTCAGGAC	451
	F5_3R	AATGCAGGTCTAGAGGACTC	
4	F5_4F	TACATGAGCATAGAAATGGGC	528
	F5_4R	TCAAACAATGATCTGGTCTCC	
5	F5_5F	CCCCAAAGCAAGAAGGTATC	488
	F5_5R	CCTTCTTGATAGGGAGTTGC	
6	F5_6F	AGGGCACAAACTACAACTGG	494
	F5_6R	TGAGGAAAGTTTGTCTGCGG	
7	F5_7F	TCTTGCCTTTTCTGGATGCC	463
	F5_7R	CCAATACATGTGTCCCCTTG	
8	F5_8F	ATGCAGGAGACAAATCAGAAG	572
	F5_8R	TTGAGAAACTGTCTCAGATCC	
9	F5_9F	AATGCTCCTGCCAAGTGATG	442
	F5_9R	AACTCCTGAAGTGAGAAGGG	
10	F5_10F	GCAATATTAATTGGTTCCAGCG	413
	F5_10R	TCTCTTGAAGGAAATGCCCC	
11	F5_11F	GGAATAGAGAATCCTTTCCC	433
	F5_11R	AAGTCTTTGGACTGGAAGTG	
12	F5_12F	AATCACTGCTTTGACACAACC	459
	F5_12R	TTGAAAGAAAAGCCTGCAGGG	
13	F5_13AF	TAGGTCACAGACAAGCAGTG	562
	F5_13AR	CTTTCTGAGGTTCTGCAAGG	
	F5_13BF	TGGCTGCAGCATTAGGAATC	549
	F5_13BR	CCAGTGTCTTGGCTAGGAAGG	
	F5_13CF	TAAGCATAAGGGACCCAAGG	568
	F5_13CR	TCTTAGAGGGTGAAAGGTCC	
	F5_13DF	AACAAGCCTGGAAAGCAGAG	501
	F5_13DR	TTCACTGAGCTCTGGAGAAG	
	F5_13EF	GACACTGGTCAGGCAAGCTG	651
	F5_13ER	TGGAGAAATGGGCATCTGAC	
	F5_13FF	AGATGCCCATTTCTCCAGAC	577
	F5_13FR	AGATCTGTCTCACCAAGGTC	
	F5_13GF	CAACCCTTTCTCTAGACCTC	520
	F5_13GR	AGTCATCTTCACTGCTCTGG	
	F5_13HF	CTTATCCAGACCTTGGTCAG	469
	F5_13HR	ATAGGGGAACCAGACTGTTC	
14	F5_14F	AGGTCATAGGAAGACTTACC	480
	F5_14R	TCACCTATAGCTCTCTTGCC	
15	F5_15F	ACTTGGGCCATATCTCACAG	502
	F5_15R	GAAATAACCCCGACTCTTCC	
16	F5_16F	GATCAATCAGAGGAAGGAGG	506
	F5_16R	GTCTCAGAAGCATCTCATGTC	
17	F5_17F	GGGAATGCAGAATCATGAGG	449
	F5_17R	TTTGGGTCTATGGGTTTGCC	
18	F5_18F	GAAAGCCTCTTGTGAAGCAG	365
	F5_18R	CAATGCAATCAGACCATGGG	
19	F5_19F	TTAAGTCAGGGCCACACAAG	357
	F5_19R	CCCAAATGGAGCTGCTTCAC	
20	F5_20F	TACAACACAGGTCCTCCAAG	339
	F5_20R	GCCTCACACTTAGTACTTGC	
21	F5_21F	AGGCAGTGTGTGACTTGTTG	355
	F5_21R	CCATATGACCCTTAGAAAGCC	
22	F5_22F	CTGGAACTGGAATTATCCCC	425
	F5_22R	CAAAGGTTTTCCTAGGAGCC	
23	F5_23F	GCCTGAGAACAGTATTTGGC	414
	F5_23R	ATACTCCTGCTTCCCAGATC	
24	F5_24F	GAGACTGTGAATCCTAAGGG	420
	F5_24R	AGAGGTGGTACATGTCACTG	
25	F5_25F	GTTTAAGGCTGCAGTGAGCC	473
	F5_25R	CTTACTTACTGGTAGCAAGGAG	

* The coding region of the 25 exons and the corresponding exon–intron boundaries were analyzed in the *F5* gene.

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
