# Peer review of "High Mutational Heterogeneity, and New Mutations in the Human Coagulation Factor V Gene. Future Perspectives for Factor V Deficiency Using Recombinant and Advanced Therapies"

_ijms, 2021, doi:10.3390/ijms22189705_

Round 1

Reviewer 1 Report

The authors present a comprehensive overview of two patients with severe FV deficiency with detailed genetic analysis and an interesting discussion regarding its homology to FVIII and an update on potential new therapies. The authors are requested to address the following questions:

  1. Page 1, line 32: The authors state that the only treatment available is the use of FFP but do not mention administration of plt concentrates which also can be utilized.  This should be added and referenced.  This statement is again repeated on page 4 lines 146-147, and on page 17 lines 551-2; in addition on page 17 the manuscript references the use of concentrates of which none are available. This should be clarified.
  2. Page 3, line115 states that "Homozygous FV deficiency.." which i believe should state that homozygous and compound heterozygous FV deficiency.
  3. Page 4, first paragraph: The authors state that the disease extremely prevalent in Caucasian countries which are home to 67% of affected patients. Is this perhaps due to bias in ability to evaluate and detect patients as opposed to countries with limited health resources?
  4. Page 4, second paragraph: The authors state that platelet derived FV is the best predictor of the severity of the condition.  This statement appears to contradict statements made in the following paragraph about how plasma levels of FV determine the severity of the condition in terms of risk of bleeding episodes. This should be clarified. In addition, on page 15, the authors state that there is not a clear association between the plasma levels of FV and the clinical expression of the disease – all these statement should be clarified and made consistent.
  5. Page 4, 4th paragraph does not mention the rare occurrence of inhibitors which should be added. This paragraph also refers to use of "Local" antifibrinolytics; I believe that local and systemic administration of antifibrinolytics is more accurate.
  6. Page 5, Case 1: The authors state that the patient "was" a Caucasian 14-year-old.." which may imply the patient is deceased.  I believe this should be changed to the word "is".
  7. Page 5, Case 1: The authors state the "other clotting factors exhibited normal levels."  The table on page 7 appears to state that the "PA" denoted as prothrombin activity" is in fact decreased in this case as well as the second case.  This requires clarification.  If i interpret prothrombin activity meaning Factor II, then the prior statement is incorrect or the issue requires clarification. On page 6 in case 2 the prothrombin activity is reported as 10.7% which is what is reported in the table. This requires further discussion as well.
  8. Page 7, line 285 Thymine is misspelled and should be corrected.
  9. Page 13, third paragraph: The authors state that some mutations identified in this study should result in a beneficial effect.  Is beneficial the correct word or do they mean neutral?
  10. Page 14, 4th paragraph: The authors state that both FV deficiency cases presented exhibit close consanguinity. I believe this is in an incorrect use of the term. The second case results from consanguinity and the first case from marriage from two endogamous communities. Please clarify.
  11. Page 15, 5th paragraph: The authors state that in Indiana consanguineous marriages account for up to 60% of marriages which is incorrect. Do the authors mean that in cases of FV deficiency this is the case?
  12. Page 15, line 472: the authors use the word “assiduous” which is an incorrect usage/meaning. In addition on page 16line 548 the word “palliative” is use which is an incorrect usage.   The same word is used on page 18line 610 which is incorrect. Do the authors mean “replacement”? Page 16, line 509 uses “y” I believe instead of the word “and”.

Reviewer 2 Report

This study describes two patients with congenital factor V deficiency resulting in expression of <1% normal factor activity. Extensive sequencing of the patients’ and first degree relatives’ factor V genes was performed and identified several mutations including some that were benign and described previously. One patient was homozygous for a deletion of a thymine residue at c.2862 leading to a frameshift and introduction of a premature stop codon resulting in production of a truncated protein. This mutation was previously described in 2002. A second patient was a double heterozygote and with nonsense mutations of the codons encoding for Arg740 and Trp1093 inherited from her mother and father, respectively, and resulting in the introduction of premature stop codons and production of a truncated protein. Mutation of the codon at Arg740 was first described in 1998, while mutation at Trp1093 has not been described previously. The authors have obtained a patent to use CRISPR/Cas9 technology to treat bleeding disorders resulting from this mutation.

While the results presented in the study appear sound and support the conclusions made, the manuscript, and in particular, the introduction and discussion are not well written, contains some flaws and needs some focusing as detailed below.

  1. A major flaw is the repeated and exclusive use of factor V and not factor Va as the active, non-enzymatic cofactor. Factor V is the procofactor, expresses NO procoagulant activity and must be activated to factor Va for function. Alluding to factor V as the active cofactor is incorrect.
  2. The authors indicate that factor V plays a key role in blood coagulation due to its procoagulant and anticoagulant properties in the very first sentences of the abstract and the introduction yet focus solely on is role on procoagulation in subsequent sentences. In addition, no primary references in support of the statement that factor V functions as an anticoagulant are provided.
  3. Likewise, the importance of the platelet form of factor V in maintaining normal hemostasis is alluded to in a single sentence in the introduction. However, its importance is understated, and its potential relevance to treatment is not noted. As a result of this omission, the research and expertise of an entire laboratory is not included.   
  4. The authors also state in the introduction that factor V “acts as a co-receptor that allosterically modifies…”. Again, no reference is provided in support of this statement.
  5. Figure 7 and the ensuing discussions of factor VIII and its structural and functional homology with factor V are unnecessary and outside the scope of the manuscript.
  6. Similarly, the discussion (start bottom page 14) of consanguineous marriages is outside the scope of this manuscript.  
  7. I would prefer to see a discussion on the use of CRISPR/Cas9 to treat disease caused by gene mutations rather than a general discussion of “advanced gene and cell therapies.”
  8. Please edit for English usage and grammar.
